# Optimal Treatment of 6-Dimethylaminopurine Enhances the In Vivo Development of Canine Embryos by Rapid Initiation of DNA Synthesis

**DOI:** 10.3390/ijms22147757

**Published:** 2021-07-20

**Authors:** Hyun Ju Oh, Byeong Chun Lee, Min Kyu Kim

**Affiliations:** 1Department of Theriogenology and Biotechnology, College of Veterinary Medicine, Seoul National University, 1 Gwanak-ro, Gwanak-gu, Seoul 08826, Korea; newborn52020@gmail.com; 2Division of Animal & Dairy Science, College of Agriculture and Life Science, Chungnam National University, Daejeon 34134, Korea

**Keywords:** dog, SCNT, DMAP, parthenogenesis, BrdU incorporation

## Abstract

Artificial activation of oocytes is an important step for successful parthenogenesis and somatic cell nuclear transfer (SCNT). Here, we investigated the initiation of DNA synthesis and in vivo development of canine PA embryos and cloned embryos produced by treatment with 1.9 mM 6-dimethylaminopurine (6-DMAP) for different lengths of time. For experiments, oocytes for parthenogenesis and SCNT oocytes were cultured for 4 min in 10 μM calcium ionophore, and then divided into 2 groups: (1) culture for 2 h in 6-DMAP (DMAP-2h group); (2) culture for 4 h in DMAP (DMAP-4h group). DNA synthesis was clearly detected in all parthenogenetic (PA) embryos and cloned embryos incorporated BrdU 4 h after activation in DMAP-2h and DMAP-4h groups. In vivo development of canine parthenogenetic fetuses was observed after embryo transfer and the implantation rates of PA embryos in DMAP-2h were 34%, which was significantly higher than those in DMAP-4h (6.5%, *p* < 0.05). However, in SCNT, there was no significant difference in pregnancy rate (DMAP-2h: 41.6% vs. DMAP-4h: 33.3%) and implantation rates (DMAP-2h: 4.94% vs. DMAP-4h: 3.19%) between DMAP-2h and DMAP-4h. In conclusion, the use of DMAP-2h for canine oocyte activation may be ideal for the in vivo development of PA zygotes, but it was not more effective in in vivo development of canine reconstructed SCNT oocytes. The present study demonstrated that DMAP-2h treatment on activation of canine parthenogenesis and SCNT could effectively induce the onset of DNA synthesis during the first cell cycle.

## 1. Introduction

Successful somatic cell nuclear transfer (SCNT) using adult somatic cells has been reported in various mammals [1,2,3,4] and has been a useful tool for producing disease models or for bio-resource purposes. This technique is affected by a variety of factors such as oocyte status, donor cell cycle, oocyte activation, and reprogramming. Although still unknown factors remain, oocyte activation has been implicated as a key factor for successful SCNT [5].

During fertilization, activation of MII oocytes is necessary for the resumption of meiotic arrest and initiation of embryo development. Oocyte activation is triggered by sperm-induced intracellular calcium oscillations [6]. This Ca^2+^ signaling causes the inactivation of maturation-promoting factor (MPF) present at high levels in MII oocytes [7]. In SCNT, cell cycle coordination between the ploidy of somatic nuclei and recipient oocyte cytoplasm is important to maintain DNA integrity and ploidy [8]. The appropriate activation method could be helpful to maintain a diploid nucleus after the transfer of a donor cell into an enucleated oocyte. The DNA status of reconstructed oocytes can be controlled by polar body extrusion or retention after activation. For this, species-specific activation protocols have been invented for producing cloned embryos.

In most mammals, oocyte activation is accomplished with calcium ionophore alone or when followed by treatment with 6-dimethylaminopurine (6-DMAP), a protein phosphorylation inhibitor that elevates the level of intracellular calcium in the oocyte and inhibits second polar body extrusion [9,10,11,12]. Inadequate or erroneous duration of treatment with 6-DMAP would cripple chromosome remodeling and subsequent development [12]. Previous studies have also shown that optimal 6-DMAP treatment (e.g., 2.5 mM 6-DMAP for 4 h in bovine and sheep, 2 mM 6-DMAP for 4 h in equine) may enhance the oocyte’s entry into the S phase of interphase by reducing the time spent in the G1 phase of the cell cycle, resulting in premature DNA synthesis [11,12,13,14]. Culturing in 10 μM ionomycin for 4 min and then 1.9 mM 6-DMAP treatment for 4 h is used for activation of canine reconstructed oocytes and parthenogenetic (PA) activation [4,15].

In a previous study which the producing intergeneric canine cloned embryos using bovine recipient oocytes, the various duration of 6-DMAP exposure affected cleavage rate on in vitro development, but not on blastocyst formation [16]. To our knowledge, there is no report on the duration of 6-DMAP exposure in in vitro development of cloned embryos by canine SCNT. Until now, no information has been available on events such as pronuclear formation or DNA synthesis occurring in reconstructed canine oocytes after activation. A thymidine analog that is incorporated into DNA, 5-Bromo-2′-deoxyuridine (BrdU), is routinely and extensively used to measure DNA synthesis and to label dividing cells. Moreover, BrdU has been used to detect DNA synthesis of SCNT and PA embryos [17,18,19,20,21]. Thus, DNA synthesis in canine embryos was evaluated with the incorporation of BrdU in this study.

The present study investigated the effect on DNA synthesis initiation in canine parthenogenetic zygotes and reconstructed SCNT oocytes after exposure to 6-DMAP for different durations and the association between DNA synthesis initiation and embryonic development in vivo.

## 2. Results

### 2.1. Effect of Activation Treatment on DNA Synthesis of Parthenogenetic Zygotes

The effect of activation treatment on DNA synthesis and in vivo development of PA zygotes was investigated. In the DMAP-2h group, DNA synthesis was initiated in all PA zygotes at 2 hpa (Table 1). However, in the DMAP-4h group, DNA synthesis in 90% of PA zygotes started at 2 h after activation (Table 1). No differences were observed in DNA synthesis at 4 h and 8 h after activation.

### 2.2. Effect of Activation Treatment on In Vivo Development of Parthenogenetic Zygotes

For in vivo development of PA zygotes, there was no significant difference in the pregnancy rate between the two groups (DMAP-2h 75% vs. DMAP-4h 66.7%, Table 2). However, the implantation rates were significantly higher in DMAP-2h (34%) compared to DMAP-4h (6.5%, *p* < 0.05, Table 2). At 26 days after ET, recovered PA embryos had developed to the stage of limb-bud formation, but a few of them showed small and degenerating structures (Figure 1).

### 2.3. Effect of Activation Treatment on DNA Synthesis of Reconstructed Oocytes Produced by SCNT

DNA synthesis in canine reconstructed oocytes following activation was observed by BrdU incorporation (Appendix A). The reconstructed oocytes of DMAP-2h group showed 90% BrdU incorporation at 2 hpa, while those of DMAP-4h group showed 77.7% BrdU incorporation at 2 hpa (*p* < 0.05, Table 3). BrdU incorporation was detected in all reconstructed oocytes of both groups after 4 hpa and 8 hpa (Table 3).

### 2.4. Effect of Activation Treatment on In Vivo Development of Canine Reconstructed Oocytes by SCNT

In results of in vivo development of reconstructed SCNT oocytes, there was no significant difference in pregnancy rate (DMAP-2h 41.6% vs. DMAP-4h 33.3%) and implantation rate (DMAP-2h 4.9% vs. DMAP-4h 3.7%) between the two groups (Table 4). Five pregnant females from the DMAP-2h group delivered nine live puppies, and four pregnant females from the DMAP-4h group delivered six live puppies by Cesarean section (Figure 2).

## 3. Discussion

Information on nuclear remodeling along with changes in DNA synthesis timing and pattern during the first cell cycle is important for improving the efficiency of SCNT. In this study, we observed DNA synthesis during the first cell cycle and in vivo development of embryos after using two activation protocols for canine parthenogenesis and SCNT. Several studies have been performed to study activation protocols for dog oocytes [15,16], however, there are no reports that systematically observed DNA synthesis timing and pattern during the first cell cycle after activation of canine oocytes. The results of this study verified that (1) DMAP exposure time during activation had an effect on DNA synthesis in the nucleus of canine PA zygotes and reconstructed SCNT oocytes, and (2) the implantation rates of canine PA zygotes derived from the DMAP-2h group were improved.

Although the factors that control the timing of the first cell division are unclear, the onset of DNA synthesis has been recognized as a critical step in the initiation of early embryo preimplantation development since it occurs immediately after pronuclear formation [20,22]. In bovine PA embryos, when high pronuclear formation rates are observed (>95%), DNA synthesis starts in about 85% of the oocytes 2 h after activation [23]. In the present study, DNA synthesis was observed in more than 70% of the oocytes 2 h after activation in both groups. DNA synthesis was evident in all one-cell stage nuclei of PA embryos and cloned embryos in both DMAP-2h and DMAP-4h groups at 4 hpa. In other species, 6-DMAP enhanced the speed of pronuclear formation, suppressed polar body extrusion, and increased cell cycle progression of calcium activated PA embryos or reconstructed oocytes [24,25,26]. It may be that 6-DMAP enhance the oocyte’s entry into S phase by reducing the time spent in the G1 phase of the cell cycle, resulting in premature DNA synthesis [14,21,27]. In this study, DNA synthesis, which indicates pronuclear formation, was confirmed faster and at higher levels in PA zygotes and reconstructed SCNT oocytes activated by DMAP-2h compared with DMAP-4h group.

With this in mind, we investigated whether faster DNA synthesis affects the in vivo developmental competence of canine PA zygotes or reconstructed SCNT oocytes. Interestingly, the number of fetuses from PA zygotes obtained by DMAP-2h treatment was significantly (*p* < 0.05) higher compared to the DMAP-4h treatment. The reconstructed SCNT oocytes of the DMAP-2h group showed a statistical difference in rapid DNA synthesis, and only numerically high results were confirmed in the pregnancy rate and implantation rate. Cattle embryos that initiated rapid DNA synthesis after removal from the activation medium resulted in sustained condensing chromatin [28], with high development to the blastocyst stage also seen [29]. Similarly, in human in vitro fertilization, the timing of the first cell cycle (initiation of DNA synthesis) appeared to be an important characteristic of the preimplantation embryo [30]. Early DNA synthesis occurs the pronuclear membrane breakdown and induces early embryo cleavage. Early zygotic cleavage has also been reported to result in a higher blastocyst formation rate [31,32]. In the present study, we showed that the rapid initiation of DNA synthesis influenced the subsequent in vivo development of the zygotes.

In conclusion, the results of this study demonstrated that DMAP-2h treatment to produce canine PA and SCNT zygotes could effectively induce the onset of DNA synthesis during the first cell cycle. An important finding of this study is that DNA synthesis of the first cell cycle was initiated 2 h after activation in canine PA and cloned embryos, and that rapid DNA synthesis effectively regulates the in vivo development of canine PA zygotes. Since the dog in vitro culture system is not well established, we recommend the 2h 6-DMAP treatment that can reduce the in vitro culture time in canine SCNT activation procedure.

## 4. Materials and Methods

### 4.1. Chemicals

Chemicals were purchased from Sigma Chemical Co. (St. Louis, MO, USA) unless otherwise stated.

### 4.2. Animals

A total of 96 mixed large-breed female dogs (Canis familiaris) between 1 and 5 years of age were used as oocyte donors and embryo transfer (ET) recipients in this study. All animal procedures were performed in accordance with recommendations described in “The Guide for the Care and Use of Laboratory Animals” published by the Institutional Animal Care and Use Committee (IACUC) of Seoul National University (Approval number; SNU-160602-14-1, Approval date; 8 May 2017).

### 4.3. Preparation of Donor Cells

Canine fibroblasts were obtained by skin biopsy cultures from 2 7-year-old beagles. Small pieces of ear skin tissue were washed and minced in phosphate-buffered saline (PBS; Invitrogen, Carlsbad, CA, USA). The minced tissues were cultured in Dulbecco’s modified Eagle’s medium (DMEM; Invitrogen) supplemented with 10% (*v*/*v*) fetal bovine serum (FBS; Invitrogen) at 38.5 °C in a humidified atmosphere of 5% CO_2_ and 95% air. A fibroblast monolayer from the tissue explants was established after 10 days. At passage zero or one, the cells were cryopreserved in 10% dimethyl sulfoxide (DMSO) and stored in liquid nitrogen. The cells from passage numbers 2 to 7 were used as donor cells for SCNT. Prior to SCNT, cells were thawed, cultured for 3 to 4 days until they were confluent, and retrieved from the monolayer by trypsinization.

### 4.4. Collection of In Vivo Matured Oocytes

The day of ovulation was considered as the day when serum progesterone concentration reached 4.0 to 7.5 ng/mL [4,33]. Serum progesterone concentration was measured with a Immulite 1000 (Siemens Healthcare Diagnostics Inc., Flanders, NJ, USA) using the CLEIA method. The ovulated oocytes were collected by aseptic surgical procedures approximately 72 h after ovulation [4,34]. Briefly, after a flushing needle was inserted into the opening of the infundibulum and tied in with a ligature, an intravenous catheter was inserted into the caudal portion of the oviduct. A 10 mL syringe with Hepes-buffered TCM (Tissue Culture Medium)-199 medium supplemented with 10% (*v*/*v*) FBS (washing medium) was introduced into the intravenous catheter of the oviduct and the flushed medium containing oocytes was collected from the flushing needle. The collected in vivo matured oocytes that were collected were transported to the laboratory within 5 min in the washing medium, held at 38.5 °C.

### 4.5. Parthenogenetic Activation

Cumulus cells from the in vivo matured oocytes were removed by pipetting 0.1% (*v*/*v*) hyaluronidase in the washing medium. For parthenogenetic activation, denuded oocytes were cultured for 4 min in washing medium supplemented with 10 μM calcium ionophore, and then divided into 2 groups for culturing in modified synthetic oviduct fluid (mSOF) medium supplemented with 1.9 mM 6-DMAP. The DMAP-2h group was cultured for 2 h in 6-DMAP, and the DMAP-4h group was cultured for 4 h in 6-DMAP. The PA zygotes were used in 2 experiments: (1) observing DNA synthesis by BrdU incorporation at 2, 4, and 8 h post-DMAP treatment (hpd), and (2) confirming fetus formation after ET. After DMAP treatment, the PA zygotes were immediately transferred surgically to the oviducts of recipients in which estrus was naturally synchronized as previously described [35,36]. Synchronous recipients (where ovulation on the same day as the oocyte donor) were identified by monitoring serum progesterone concentration. A total of 75 PA zygotes derived from 9 oocyte donor dogs were transferred to 7 surrogate recipients (42 PA zygotes into 4 recipients of DMAP-2h and 33 PA zygotes into 3 recipients of DMAP-4h). Fetal retrieval day and fetal formation criteria were determined based on previous studies [15,37,38]. The parthenogenetic fetuses were collected from recipients 28 days after ET [15] and fetal formation was assessed based on limb-bud formation [15,37,38]. The evaluation of the in vivo development of PA embryos were determined by pregnancy and implantation rates. Pregnancy rates were determined by dividing the number of pregnant dogs by the total number of recipients and implantation rates were determined by dividing the number of fetuses formed on day 28 post-ET by the total number of transferred PA embryos.

### 4.6. Somatic Cell Nuclear Transfer

For canine SCNT, in vivo matured oocytes were enucleated, then donor cells were microinjected into the perivitelline space and fused by electrical stimulation. The fused couplets were cultured for 4 min in 10 μM calcium ionophore, and then divided into 2 groups with the addition of 1.9 mM 6-DMAP. The DMAP-2h group was cultured for 2 h in 6-DMAP, and the DMAP-4h group was cultured for 4 h in 6-DMAP. Activated cloned embryos were subjected to 2 analyses: (1) pronuclear formation was observed by BrdU incorporation at 2, 4, and 8 h post-activation (hpa), and (2) fetus formation, and pregnancy efficiency after ET into naturally synchronous recipients was determined. Reconstructed SCNT oocytes were immediately transferred to a recipient after activation. A total of 370 cloned embryos derived from 43 oocyte donor dogs were transferred to 24 surrogate recipients (182 cloned embryos into 12 recipients for DMAP-2h and 188 cloned embryos into 12 recipients for DMAP-4h). Pregnancy diagnosis was performed by ultrasonography on day 26 after ET. The pregnancy rates were determined by dividing the number of pregnant females on day 26 post-ET by the total number of recipients. The implantation rates were determined by dividing the number of puppies born by the total number of transferred cloned embryos.

### 4.7. Staining with 5-Bromo-20-Deoxyuridine Staining for Evaluation of DNA Synthesis in Embryos

Synthesis of DNA was determined by the incorporation of BrdU [39,40,41]. In order to examine DNA synthesis in the DMAP-2h and DMAP-4h groups, parthenogenetic and SCNT embryos were cultured in mSOF medium at 38.5 °C in a humidified atmosphere of 5% CO_2_. The experimental procedure was similar to that described previously [39,40,41]. Briefly, embryos were transferred into a culture medium containing 10 μM BrdU 1 h prior to the juncture of 2, 4, and 8 h post-activation (hpa). The embryos were incubated for 1 h at 38.5 °C in a humidified atmosphere of 5% CO_2_, then were fixed in 4% paraformaldehyde. Permeabilization was carried out in 0.1% Triton X-100 for 15 min at room temperature. After permeabilization, embryos were incubated in mouse anti-BrdU (1:10) solution for 30 min at 38.5 °C. Subsequently, embryos were incubated with fluorescein isothiocyanate (FITC)-conjugated rabbit anti-mouse (1:10) antiserum for 30 min at 38.5 °C and finally counterstained with DAPI for 5 min at room temperature. For visualization, embryos were mounted on glass slides and observed under a fluorescence microscope. DNA synthesis was considered to have occurred (positive) when green fluorescence could be observed under UV light.

### 4.8. Statistical Analysis

The experiments examining parthenogenetic zygotes and reconstructed SCNT oocytes, and subsequent in vivo development were repeated at least 3 times. Statistical analyses were performed using GraphPad Prism version 5 (GraphPad Software, San Diego, CA, USA). A Fisher’s exact test was used for BrdU incorporation rate, pregnancy rate, implantation rate, and cloned puppy birth rate between the DMAP-2h and DMAP-4h groups. The significance level was *p* < 0.05.

## Figures and Tables

**Figure 1 ijms-22-07757-f001:**
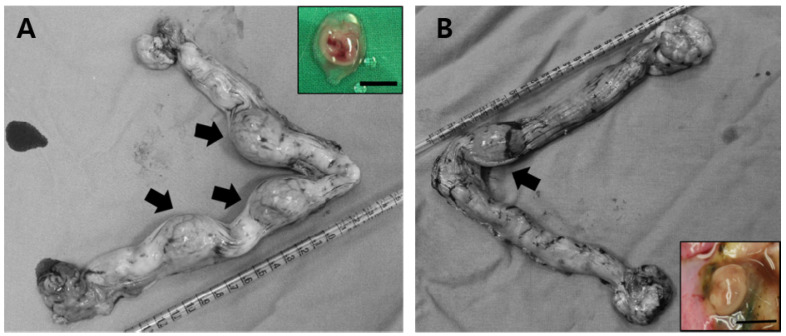
In vivo development of 26-day fetuses following PA. Implantations and dissected fetuses (inner boxes) derived from the DMAP-2h (**A**) and DMAP-4h (**B**) groups at 26 days. Arrows indicate the placental sacs on day 26. All scale bars represent 10 mm.

**Figure 2 ijms-22-07757-f002:**
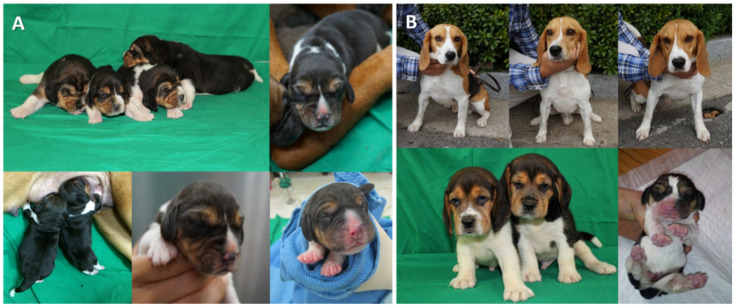
Cloned puppies produced by SCNT. (**A**) Nine cloned puppies derived from the DMAP-2h group. (**B**) Six cloned puppies derived from the DMAP-4h group.

**Table 1 ijms-22-07757-t001:** BrdU incorporation in canine parthenogenetic zygotes activated by the different duration of 6-DMAP exposure.

Treatment	Timing (hpa)	No. Flushed Oocytes	No. PA Zygotes	No. PA Zygotes Labeled with BrdU (%)
DMAP-2h	2	10	10	10 (100) ^a^
	4	12	12	12 (100)
	8	12	12	12 (100)
DMAP-4h	2	10	10	9 (90) ^b^
	4	12	12	12 (100)
	8	11	11	11 (100)

hpa, hours post-activation. ^a,b^ means a significant difference in parthenogenetic zygotes labeled with BrdU between the DMAP-2h and DMAP-4h (*p* < 0.05).

**Table 2 ijms-22-07757-t002:** Effect of different durations of 6-DMAP exposure on in vivo development of canine parthenogenetic zygotes.

Treatment	Trial No.	No. Flushed Oocytes	No. Transferred PA Zygotes	Pregnancy Rate ^1^	No. Implantations (%) ^2^
DMAP-2h	1	11	11	+	3
	2	10	10	+	8
	3	7	7	+	2
	4	14	14	-	0
Total		42	42	75%	13 (34) ^a^
DMAP-4h	1	12	12	+	1
	2	12	12	-	0
	3	9	9	+	1
Total		33	33	66.7%	2 (6.5) ^b^

^1^ Percentage was calculated from the number of recipients. ^2^ Percentage was calculated from the number of parthenogenetic zygotes transferred. ^a,b^ means a significant difference in implantation rate between the DMAP-2h and DMAP-4h (*p* < 0.05).

**Table 3 ijms-22-07757-t003:** BrdU incorporation in canine reconstructed oocytes activated by different incubation time with 6-DMAP.

Treatment	Timing (hpa)	No. Flushed Oocytes	No. Reconstructed Oocytes	No. Oocytes Labeled with BrdU (%)
DMAP-2h	2	26	20	18 (90) ^a^
	4	16	12	12 (100)
	8	18	14	14 (100)
DMAP-4h	2	23	18	14 (77.7) ^b^
	4	15	12	12 (100)
	8	18	14	14 (100)

hpa, hours post-activation. ^a,b^ means significant difference in reconstructed oocytes labeled with BrdU between DMAP-2h and DMAP-4h.

**Table 4 ijms-22-07757-t004:** Effects of different durations of 6-DMAP exposure on the in vivo development of reconstructed oocytes by SCNT.

Treatment	No. Flushed Oocytes	No. Reconstructed Oocytes (Transferred)	No. Recipients	No. Pregnant (%) ^1^	No. Cloned Puppies (%) ^2^
DMAP-2h	232	182	12	5 (41.6)	9 (4.9)
DMAP-4h	242	188	12	4 (33.3)	6 (3.2)

^1^ Percentage was calculated from the number of recipients. ^2^ Percentage was calculated from the number of reconstructed SCNT oocytes transferred.

## Data Availability

All data used to support the findings of this study are included within the article. The analyzed data during the current study are available from the corresponding author upon reasonable request.

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
