# Peer review of "Optimal Treatment of 6-Dimethylaminopurine Enhances the In Vivo Development of Canine Embryos by Rapid Initiation of DNA Synthesis"

_ijms, 2021, doi:10.3390/ijms22147757_

Round 1

Reviewer 1 Report

The current paper is written in proper English grammar with a logical sequence and describes well the methods.

The study aimed to investigate the effect of the treatment of canine oocytes with 6-DMAP on oocyte activation and PA or SCNT success. The authors succeed in explaining their hypothesis based on the review presented in the introduction section.  The material and methods sections need to be improved providing more information about the number of oocytes. The embryo development rates. The statistics are poorly written and need to be improved. Despite the paper be related to an elegant technique of interesting biotechnology, a simple design was presented. Therefore, the presented conclusions may be helpful in further research. Finally, I do recommend the paper for publishing in IJMS after Minor Revisions.

Minor Revisions

Abstract

Line 25- Please clarify the sentence “between DMAP-2h and DMAP-2h.”  

Line 26- It is not clear how the authors evaluated the embryo. The abstract suggests that 6-DAMP was less effective in establish pregnancies. Nothing was presented regarding embryo quality or percentage of development. The conclusions need to be drawn according to the presented methods.

Results

Line 92- At table 2, please add the information about the number of oocytes, blastocyst development rate

Line 107- At table 3, add information about the number of treated oocytes.

Methods

Line 263- The statistics are poorly written and does not indicate the number of treated oocyte.

Tables 1, 2, and 3 are not fully auto-explained. The table titles must have all info. 

Author Response

Thanks for the reviewer's comments. Every detail pointed out by reviewers contributed greatly to improving our manuscript. The manuscript has been revised, and for the response to the reviewer's comments, please check the attachment.

Reviewer 2 Report

The manuscript presents the data about the use of 6-dimethylaminopurine for 2 or 4 hours to enhance formation and in vivo development of  parthenogenetic and cloned canine embryos. Activation of oocytes is a key step of those procedures and the results are important for the development of SCNT in the dog.

The manuscript is well written and properly structured. Material and methods part could have been elaborated more, e.g. what was the method of progesterone assessment? When were the cloned ‘embryos’ transferred – immediately, as presumptive zygotes or were they cultured before the transfer?  

The aim of the study is not clearly justified – comparison of different incubation time with 6-DMAP has been studied previously in the dog and in other species and best results were obtained with 4 hours incubation. Why shortened incubation time was investigated?

There are other minor remarks:

  • The Authors are using the term ‘embryo’ when describing oocytes few hours after activation. The term ‘embryo’ is used after the first cleavage, before the first cleavage it is a zygote. If I understand correctly, (pseudo)pronucleus formation was not checked – so they were even presumptive zygotes. Please correct it through the manuscript
  • Table 2 – there is little mess in superscripts in “pregnancy rate” column
  • Table 3 and 4 – the term ‘parthenotes’ is typically used for parthenogenetically developed embryos, here SCNT zygotes /embryos should be used
  • Line 113 – it is stated that ‘six pregnant females from the DMAP-4h group’ whereas Table 4 shows four pregnant females in this group
  • Line 185 – what do you mean by two kind of fibroblasts?

I believe that after a minor revision this manuscript will be suitable for publication in the IJMR.

Author Response

(The authors gave the same response as above.)
